# Diversity and Composition of Soil Acidobacterial Communities in Different Temperate Forest Types of Northeast China

**DOI:** 10.3390/microorganisms12050963

**Published:** 2024-05-10

**Authors:** Feng Jiao, Lili Qian, Jinhua Wu, Dongdong Zhang, Junying Zhang, Mingyu Wang, Xin Sui, Xianbang Zhang

**Affiliations:** 1College of Agriculture, Heilongjiang Bayi Agricultural University, Daqing 163319, China; jiaofeng_1980@126.com (F.J.); qianlili2252588806@byau.edu.cn (L.Q.); wujinhua1977@163.com (J.W.); a230224127@126.com (D.Z.); byndzjy08315@163.com (J.Z.); 2Engineering Research Center of Agricultural Microbiology Technology, Ministry of Education & Heilongjiang Provincial Key Laboratory of Ecological Restoration and Resource Utilization for Cold Region & Key Laboratory of Microbiology, College of Heilongjiang Province & School of Life Sciences, Heilongjiang University, Harbin 150080, China; wmy022234@163.com; 3Heilongjiang Zhongyangzhan Black—Billed Capercaillie National Nature Reserve Service Center, Nenjiang 161400, China; zh_xi_ba@163.com

**Keywords:** Acidobacteria, *Larix gmelinii*, soil organic carbon, diversity, composition

## Abstract

To gain an in-depth understanding of the diversity and composition of soil Acidobacteria in five different forest types in typical temperate forest ecosystems and to explore their relationship with soil nutrients. The diversity of soil Acidobacteria was determined by high-throughput sequencing technology. Soil Acidobacteria’s alpha-diversity index and soil nutrient content differed significantly among different forest types. β-diversity and the composition of soil Acidobacteria also varied across forest types. Acidobacterial genera, such as *Acidobacteria*_Gp1, *Acidobacteria*_Gp4, and *Acidobacteria*_Gp17, play key roles in different forests. The RDA analyses pointed out that the soil pH, available nitrogen (AN), carbon to nitrogen (C/N) ratio, available phosphorus (AP), total carbon (TC), and total phosphorus (TP) were significant factors affecting soil Acidobacteria in different forest types. In this study, the diversity and composition of soil Acidobacteria under different forest types in a temperate forest ecosystem were analyzed, revealing the complex relationship between them and soil physicochemical properties. These findings not only enhance our understanding of soil microbial ecology but also provide important guidance for ecological conservation and restoration strategies for temperate forest ecosystems.

## 1. Introduction

Forests are the mainstay of terrestrial ecosystems, which play important roles in purifying the air, regulating the climate, conserving water resources, maintaining biodiversity, and also providing a large amount of material resources for human survival [1,2]. However, due to global climate changes and human activities, forests are being destroyed, which has a serious impact on the balance and stability of the entire ecosystem. At the same time, the destruction of forests also leads to increased soil erosion and reduced soil fertility, affecting the sustainable development of ecosystems and posing a threat to the survival and development of humankind [3,4]. At the same time, the protection of forest ecosystems has become a global issue. Governments and international organizations are cooperating extensively in the field of forest protection and are jointly addressing the challenges of global climate change and biodiversity conservation [5,6]. Therefore, the protection of forest ecosystems has received more and more attention from researchers [7,8]. Therefore, we conducted a detailed study of the different forest ecosystems in Zhongyangzhan Black-billed Capercaillie Nature Reserve in Heilongjiang Province.

Acidobacteria, as an important component of the soil bacterial community, is one of the most abundant bacterial taxa in soil, characterized by a wide range of metabolic and genetic functions, which can be found in a variety of habitats [9]. It is well known that soil bacterial communities are highly susceptible to changes in the surrounding environment [10], and Acidobacteria are no exception [11]. Changes in external environmental conditions affect the outcomes and functions of soil Acidobacterial communities [12]. In particular, changes in soil Acidobacterial communities were more pronounced under different forest ecosystems. For example, Catao et al. previously reported that the diversity of soil Acidobacteria varied among forest types and that the top-ranked *Acidobacteria* genera differed in abundance [13]. This indicated the importance of studying Acidobacteria in forest ecosystems.

It is worth noting that soil physico-chemical properties seem to strongly influence the species composition of Acidobacteria. Campanharo et al. [14,15] found that the effect of pH on the growth of members of Acidobacteria was significant, with the strongest effect of these conditions at pH = 5. Sait et al. [16] found that significantly more Acidobacteria colonies formed at pH = 5.5 than at pH = 7. All these results suggest that Acidobacteria preferred an acidic environment. Changes in other soil nutrients also significantly affect changes in Acidobacteria. For example, Li et al. [17] found a strong positive effect of total nitrogen, available nitrogen, and soil organic carbon on Acidobacteria. Given the above background, this study not only investigated the changes in the diversity and composition of soil Acidobacterial communities under different forest types but also analyzed in depth the effects of soil physicochemical properties on soil Acidobacterial communities under different forest types.

The Heilongjiang Zhongyangzhan Black-billed Capercaillie Nature Reserve is located at the intersection area between Daxing’anling and Xiaoxing’anling mountains in temperate Northern China. The nature reserve contributes significantly to ecosystem stability and ecosystem services. We conducted a comparative study of soil Acidobacterial communities in five different forest vegetation types [(*Betula dahurica* (BD) forest, *Betula platyphylla* (BP) forest, *Quercus mongolica* (QM) forest, *Q. mongolica* and *L. gmelinii* mixed (LGQM) forest, and *Larix gmelinii* (LG) forest)] in the nature reserve. The diversities and compositions of soil Acidobacterial communities were measured by 16S rRNA genes (V3–V4) using Illumina MiSeq technology. We hypothesized that the diversity and composition of soil Acidobacterial community will change to some degree under different forest types. This study will provide basic data for further explaining the ecological functions of Acidobacteria in soil and also for in-depth revealing of the function of Acidobacteria in the element cycle in the process of forest ecosystems.

## 2. Materials and Methods

### 2.1. Experimental Site

The study sites are located in the Zhongyangzhan Black-billed Capercaillie Nature Reserve (126°00′–126°45′ N, 48°30′–48°50′ E), Heilongjiang Province, China (Figure 1). The region has a temperate continental climate with an average temperature of −0.4 °C, an average annual precipitation of 450–550 mm, and covers an area of 7274 ha. The dominant forest types in the region are coniferous, mixed coniferous, and broad-leaved forests. Five forest types with representatives were selected for this study, namely, *Betula dahurica* (BD) forest, *Betula platyphylla* (BP) forest, *Quercus mongolica* (QM) forest, *Q. mongolica* and *L. gmelinii* mixed (LGQM) forest, and *Larix gmelinii* (LG) forest. The main plants in the nature reserve are *Populus davidiana*, *Choseniaarbutifolia*, *Betula platyphylla*, *Larix gmelinii*, *Tilia amurensis*, *Prunus padus*, *Betuladahurica Pall*, *Salix raddeana*, *Quercus mongolica* and *Alnus mandshurica*.

### 2.2. Soil Sampling

Soil samples were collected in July 2019 under five different forest types, namely, *Betula dahurica* forest, *Betula platyphylla* forest, *Quercus mongolica* forest, *Q. mongolica* and *L. gmelinii* mixed forest, and *Larix gmelinii* forest. Soil samples were taken from three 20 m × 20 m forest plots, and soil samples from each plot were mixed. Soil samples (0~20 cm) were collected using a soil auger (5 cm in diameter and 20 cm in depth) after removing the litter layer. The collected soil samples were immediately stored at 4 °C temperature and subsequently divided into two sub-samples, one of which was air-dried, crushed, and milled pending subsequent testing of the physical and chemical properties of the soil, and the other soil sample was stored at −20°C for subsequent DNA extraction.

### 2.3. Soil Physicochemical Properties

Soil pH testing using a soil pH meter (soil-to-water ratio of 1:2.5 *w*/*v*) [18,19]. The soil–water mixture was stirred thoroughly until a homogeneous suspension was formed. The soil pH meter was switched on, the electrode was inserted into the suspension, and the pH value was recorded after the reading had stabilized. To ensure accurate data, three replicates were taken on the same sample and averaged. Soil total carbon (TC) and total nitrogen (TN) were tested by an elemental analyzer (Elementar, Langenselbold, Germany) (Take care to set the parameters of the elemental analyzer, including combustion temperature and gas flow rate, to ensure accurate measurement conditions) [20]. Soil carbon-to-nitrogen ratio (C/N) was tested using the thermal Conductivity Cell Detector (TCD) (When using the thermal Conductivity Cell Detector (TCD), ensure that the purity and flow rate of the carrier gas meets the requirements of the instrument to avoid compromising the accuracy of the results) [21]. Measurement of soil available nitrogen (AN) was performed using a continuous flow analysis system (SKALAR SAN++) [22]. Soil total phosphorus (TP) content was measured by the HCLO_4_-H_2_SO_4_ method [23]. Soil available phosphorus (AP) content was determined by the colorimetric method upon extraction with 0.5 M NaHCO_3_ (When using 0.5 M NaHCO_3_ solution for oscillatory extraction, ensure that the temperature is constant and the duration of oscillation is sufficient to fully extract the AP) [24].

### 2.4. Soil DNA Extraction and 16S rDNA Sequencing

DNA was extracted from 0.4 g of soil samples using a soil DNA kit (OMEGA BIO TEK, Norcross, GA, USA) and subsequently assayed using the NanoDrop-1000 spectrometer (Nanodrop, Athens, GA, USA). We then used the special primers ACIDO (5′GCTCAGAATSAACGCTGG3′)/342r(5′CTGCTGCSYCCCGTAG3′) (~336 bp), which were selected to amplify the Acidobacterial region [25]. The qPCR was performed under the following conditions: 20 μL PCR reactants; 0.3 ng template DNA; 250 nM of forward and reverse primers. The amplification conditions were as follows: initial denaturation at 95 °C for 5 min; denaturation at 95 °C for 15 s; annealing and extension at 55 °C for 1 min over 30 cycles [26]. PCR was repeated 3 times for each sample, and then the resulting samples were mixed. PCR products were sequenced on an Illumina MiSeq paired-end (PE = 300) platform at Majorbio (Shanghai International Medical Zone, Shanghai, China).

### 2.5. Bioinformatics and Statistical Analysis

One-way analysis of variance (ANOVA) with Duncan’s test was used to analyze the differences in soil physical and chemical properties under five different forest types using SPSS software v26.0 [27]. Soil Acidobacterial alpha-diversity indices, such as soil Acidobacterial Richness index, Chao1 index, abundance-based cover estimation (ACE), Shannon index, and Evenness index, were calculated using QIIME2 [28]. Score plots of principal component analysis (PCA) were drawn using the “stats”, “psych”, “FactoMine”, “vegan”, “ggbiplot” packages in the R4.3.1 software (version 3.2.3) [29]. The Venn diagram was drawn using “ggvenn” and “ggsave” packages in the R software (version 3.2.3) [30,31]. The stacked plot was drawn using the “ggplot” package. Mapping of the random forest classification modeling was performed using the “RandomForest” Package in the R software (version 3.2.3) [32,33]. Kruskal–Wallis test using the pairwise_wilcox_test function from the “rstatix” package in the R software (version 3.2.3) [34,35]. Mantel test using the “Vegan” and “ggcor” packages in the software (version 3.2.3) [36,37]. A redundancy analysis (RDA) diagram was drawn by “ggplot2”, “vegan” and “ggvegan” packages in the software (version 3.2.3) [38,39]. Construct a correlation heatmap using the “corrplot” package in the R software (version 3.2.3) [40].

## 3. Results

### 3.1. Soil Properties

Soil physico-chemical properties at different forest types are shown in Table 1. Except for TP, other soil physicochemical properties differed significantly among different forest types (Table 1, *p* < 0.05). Among them, TC was the highest in *Betula platyphylla* (BP) forest soils, and C/N and AP were the highest in *Quercus mongolica* (QM) forest soils. Nevertheless, pH was lowest in *Betula dahurica* (BD) forest soils; TN was the lowest in *Q. mongolica* and *L. gmelinii* mixed (LGQM) forest soils. AN and AP were the lowest in *Larix gmelinii* (LG) forest soils.

### 3.2. Soil Acidobacterial Diversity in Five Forest Types

Significant differences were found among alpha diversity indices of Acidobacteria. The Richness, Chao1, ACE, Shannon, and Evenness indices of soil Acidobacterial community in *Betula dahurica* (BD) forest and *Betula platyphylla* (BP) forest were significantly higher than in other three forest types (*Larix gmelinii* (LG) forest, *Q. mongolica* and *L. gmelinii* mixed (LGQM) forest, and *Quercus mongolica* (QM) forest). On the contrary, the Evenness index had no significant difference in five different forest types (Table 2).

In addition, it is worth noting that the soil Acidobacterial community’s composition significantly altered according to different forest soils (Figure 2). In detail, from the scores plot of principal component analysis (PCA), the soil Acidobacterial beta diversity of *Larix gmelinii* (LG) forest differed significantly compared to the other four forest types. The soil Acidobacterial beta diversity is extremely similar in *Quercus mongolica* (QM) forest and *Q. mongolica* and *L. gmelinii* mixed (LGQM) forest. The alpha diversity of soil Acidobacteria community structure in *Betula dahurica* (BD) and *Betula platyphylla* (BP) forest soils was similar.

### 3.3. Differences in Soil Acidobacterial Composition in Five Forest Types

The Venn diagram based on the OTUs level demonstrated that OTUs differed among the five forest types (Figure 3). The total number of OTUs of soil Acidobacterial communities shared among the five forests was 595 (Figure 3). The *Betula platyphylla* (BP) forest exhibited the largest number of unique OTUs (469; Figure 3). The number of shared soil Acidobacterial OTUs was the largest for the pair *Betula dahurica* (BD) forest and *Betula platyphylla* (BP) forest (251; Figure 3). The number of shared soil Acidobacterial OTUs was the largest for the pair *Betula platyphylla* (BP) forest, *Larix gmelinii* (LG) forest, and *Quercus mongolica* (QM) forest (37; Figure 3). These OTU numbers indicated that forest type was a major factor influencing the soil Acidobacterial communities.

From the Stacked plot, under five different forests, the top-ranked dominant subgroups were *Acidobacteria*_Gp7, *Acidobacteria*_Gp6, *Acidobacteria*_Gp4, and *Acidobacteria*_Gp2 (Figure 4a). However, surprisingly, random forest classification models to predict microbial taxa that play key roles in soil Acidobacterial communities revealed different patterns (Figure 4b). Generally speaking, the results showed that the lower abundance subgroup in the soil Acidobacterial community nevertheless played an important role in the soil Acidobacterial community. However, *Acidobacteria*_Gp6, with the highest abundance, also plays an important role in the soil Acidobacterial community in five forest types.

Kruskal–Wallis test showed that different Acidobacterial subgroups had significant differences (*p* < 0.05) in five forest types (Figure 5). It is worth noting that *Acidobacteria*_Gp1 had the most significant difference (*p* < 0.001) among the five forest types compared to the other Acidobacteria genera. In addition, by comparing the abundance of different Acidobacterial subgroups in different forest types, it can be seen that *Acidobacteria*_Gp1 has the lowest abundance in *Betula platyphylla* (BP) forest. However, the number of *Acidobacteria*_Gp6 and *Acidobacteria*_Gp4 were relatively higher in the *Betula platyphylla* (BP) forest than in other forest types (Figure 4a and Figure 5). The number of *Acidobacteria*_Gp2 and *Acidobacteria*_Gp1 was the highest in the *Q. mongolica and L. gmelinii* mixed (LGQM) forest (Figure 4a and Figure 5). The number of *Acidobacteria*_Gp4 was the highest in *Larix gmelinii* (LG) forest and *Betula dahurica* (BD) forest (Figure 4a and Figure 5). The number of *Acidobacteria*_Gp2 was the highest in the *Quercus mongolica* forest (QM) (Figure 4a and Figure 5).

### 3.4. Effect of Soil Physicochemical Properties on Soil Acidobacterial Communities in Five Forest Types

As shown in Figure 6, we demonstrated the interaction between Acidobacteria and soil physicochemical properties by plotting the correlation heatmap (Figure 6). Through the heatmap diagram, there was a significant positive correlation between *Acidobacteria*_Gp1 and *Acidobacteria*_Gp3 with pH. *Acidobacteria*_Gp19 and *Acidobacteria*_Gp21 showed a significant positive correlation with TP, AP, and AN. Differently, *Acidobacteria*_Gp17, *Acidobacteria*_Gp22, *Acidobacteria*_Gp25, *Acidobacteria*_Gp4, *Acidobacteria*_Gp6, and *Acidobacteria*_Gp7 had significant negative correlation with pH. Not only that but there was also a significant negative correlation between *Acidobacteria*_Gp23 and AP.

Mantel analysis revealed the correlation between the α-diversity, composition of soil Acidobacterial communities, and soil physicochemical indicators (Figure 7). The soil Acidobacterial communities’ α-diversities and compositions were related to soil TN. In addition, the soil Acidobacterial communities’ α-diversities also correlated with soil pH.

The RDA demonstrated the relationship between soil physicochemical properties and forest types (Figure 8). In addition, the RDA model explained 96.56% of the total variance. Soil pH was positively related to the *Q. mongolica* and *L. gmelinii* mixed (LGQM) forest; soil AN and TN were positively related to the *Betula dahurica* (BD) forest. In addition, soil C/N, TC, TP, and AP were positively correlated with the *Quercus mongolica* (QM) forest but were negatively correlated with the *Betula platyphylla* (BP) forest and *Larix gmelinii* (LG) forest.

## 4. Discussion

### 4.1. Effects of Forests on Soil Acidobacterial Diversity and Composition

According to my previous comment, our results showed that soil Acidobacterial community alpha-diversity changed significantly during different forest types in Heilongjiang Zhongyangzhan Black-billed Capercaillie Nature Reserve, which is consistent with previous reports [41,42]. Soil Acidobacterial alpha-diversity was significantly higher in *Betula dahurica* (BD) forest and *Betula platyphylla* (BP) forest than in *Q. mongolica* and *L. gmelinii* mixed (LGQM) forest. In this regard, we believe that both *Betula dahurica* (BD) forest and *Betula platyphylla* (BP) forest are composed of a single plant of the genus Birch in the family Birchaceae, and that plants of the same genus have relatively close affinities, with some similarities in pathways of access to soil nutrients and occupation of ecological niches [43]. We assume that the alpha diversities of Acidobacteria, both *Betula dahurica* and *Betula platyphylla,* could be related to similar root exudates [44]. Therefore, soil Acidobacterial alpha-diversities were similar in *Betula dahurica* (BD) forest and *Betula platyphylla* (BP) forest. Single species had less effect on soil nutrient content, resulting in higher soil Acidobacterial alpha-diversities in *Betula dahurica* (BD) forest and *Betula platyphylla* (BP) forest. In contrast, the *Q. mongolica* and *L. gmelinii* mixed (LGQM) forest is composed of two different types of tree species, *Quercus mongolica* and *Larix gmelinii*, which differ in their ecological habits and can adapt to different environmental conditions [45,46], showing greater ecological diversity. The alpha-diversity of soil Acidobacteria in *Q. mongolica* and *L. gmelinii* mixed (LGQM) forest was reduced because changes in forest vegetation led to changes in soil nutrient content, and Acidobacteria are more sensitive to changes in the soil environment. Different forest types significantly impact the alpha-diversity of soil Acidobacteria, highlighting their key role in forest ecosystems and deepening our understanding of biodiversity and forest types. Changes in the alpha-diversity of soil Acidobacteria reflect global environmental impacts on forests. Protecting and restoring specific forests maintains soil Acidobacteria diversity, aids forest management, and provides a scientific basis for understanding forest responses to climate change. This is crucial for global biodiversity conservation, forest ecosystem research, and climate change studies.

In addition, PCA analyses revealed that soil Acidobacterial β-diversity in *Larix gmelinii* (LG) forest was significantly different from that under the other four forest types [(*Betula dahurica* (BD) forest, *Betula platyphylla* (BP) forest, *Quercus mongolica* (QM) forest, and *Q. mongolica* and *L. gmelinii* mixed (LGQM) forest)] (Figure 2). This is consistent with Sui et al.’s study [47], which confirmed that the soil bacterial β-diversity of *Larix gmelinii* (LG) was different from the other four forest types. β-diversity, also known as between-habitat diversity, is the dissimilarity of species composition between different habitat communities along an environmental gradient or the rate of species turnover along an environmental gradient [48,49]. In contrast, the environments of the top forests differed significantly from those of the other four secondary forest types, and it is for this reason that the β-diversity of soil Acidobacteria varied significantly under different forest types. We speculated that this might be due to variations in soil nutrient content in different forest types. There are large differences in vegetation types between the top and secondary forest types, and changes in both forest litter and plant root secretion types affect the survival, reproduction, and metabolism of other species of soil microorganisms, which, in turn, lead to differences in soil nutrient content. This makes the Acidobacteria community, which is sensitive to environmental changes, different under different forest types. This part reveals the effect of different forest types on soil Acidobacteriaβ-diversity and finds significant differences between larch forests and other secondary forest types. This finding emphasizes the importance of ecosystem diversity, demonstrates the role of different forest types in shaping Acidobacterial communities, and provides a scientific basis for forest management and restoration.

From the results of the Venn diagram, it is clear that the highest number of OTUs specific to Acidobacteria was found in the *Betula platyphylla* (BP) forest, and the lowest number of OTUs specific to Acidobacteria was found in the *Q. mongolica* and *L. gmelinii* mixed (LGQM) forest (Figure 3). It is obvious that soil Acidobacteria are differently adapted in the two forest types, *Betula platyphylla* (BP) forest and *Q. mongolica* and *L. gmelinii* mixed (LGQM) forest, and therefore, different Acidobacteria are able to survive and multiply stably. *Larixgmelinii* (Rupr.) Kuzen. of *Q. mongolica* and *L. gmelinii* mixed (LGQM) forests are hardy and soil-adapted, growing in swamps, on dry, sunny slopes, as well as on moist shady slopes. *Betula platyphylla* Sukaczev. is light-loving and adaptable [50,51] but is more commonly found in moist soil conditions and is a pioneer species in secondary forests. Differences in the adaptation of above-ground vegetation to the environment in the two forest types led to variations in soil nutrient content, which, in turn, led to different OTUs of soil Acidobacteria specific to the different forest types. The results of the Venn diagram demonstrate that Acidobacteria has different ecological adaptations under different forest types. This finding not only deepens our understanding of the interactions between Acidobacteria and the environment but also provides an important basis for biodiversity conservation, environmental monitoring, and soil health assessment.

Stacked plots indicate that *Acidobacteria*_Gp6, *Acidobacteria*_Gp4, and *Acidobacteria*_Gp2 rank high among the five different forest types (Figure 4a). In addition, the random forest model shows that *Acidobacteria*_Gp1, *Acidobacteria*_Gp4, and *Acidobacteria*_Gp17 are key in the composition of the five forest types (Figure 4b). The above results demonstrate that *Acidobacteria*_Gp4 is both a dominant genus and a key species playing an important role under the five forest types. Not only that, the Kruskal–Wallis test selected the top six Acidobacteria genera in terms of total abundance to be analyzed (Figure 5). It can be intuitively observed that *Acidobacteria*_Gp4 still occupies an important position, and the abundance of *Acidobacteria*_Gp4 is the highest in the *Betula platyphylla* (BP) forest. This is once again strong evidence of the importance of *Acidobacteria*_Gp4 in different forest types. This may be due to the fact that *Acidobacteria*_Gp4 is acidophilic and highly adaptable to its surroundings [52,53]. As a result, it is both able to survive and reproduce in large numbers in forests and plays a key role in forest composition. In addition, the key species that play important roles in the five different forest types are *Acidobacteria*_Gp1 and *Acidobacteria*_Gp17. The reasons for the key role of *Acidobacteria*_Gp1 in forests may be closely related to its unique ecological niche and physiological functions. *Acidobacteria*_Gp1 is able to decompose complex organic materials, such as cellulose and lignin, which are the main components of forest litter [54,55]. By decomposing these organic substances, *Acidobacteria*_Gp1 releases small molecules that can be utilized by other organisms, such as amino acids and sugars, and, thus, promotes the material cycle of forest ecosystems. In addition, *Acidobacteria*_Gp1 may take an advantageous position in the interaction with other soil microorganisms. Together, these factors make *Acidobacteria*_Gp1 an indispensable and important component of forest ecosystems. The key role of *Acidobacteria*_Gp17 in forests may be due to their unique adaptations to the soil environment [56]. Their ability to survive and reproduce under extreme conditions, such as acidic and oligotrophic conditions, has led to their wide distribution and high abundance in forest soils. Such adaptations may allow *Acidobacteria*_Gp17 to play a key role in maintaining ecological balance and providing ecosystem services in forest soils. Thus, *Acidobacteria*_Gp17 and *Acidobacteria*_Gp1 are the keystone species under five different forest types. Specific Acidobacteria (e.g., *Acidobacteria*_Gp4, *Acidobacteria*_Gp1, and *Acidobacteria*_Gp17) have different key roles in forest ecosystems, which deepens our understanding of soil Acidobacteria influences on forest functioning. In addition, the diversity and function of soil Acidobacteria as sensitive indicator organisms of environmental change are critical for forest health and stability and provide important ecological services through material cycling.

### 4.2. Mantel Analyses Correlation between Soil Physicochemical Properties and Soil Acidobacterial Composition in Different Forest Types

The heat map showed us the proximity between different Acidobacteria and different soil physico-chemical properties (Figure 6). Among the key species that played an important role, *Acidobacteria*_Gp1 showed a significant positive correlation with pH, and *Acidobacteria*_Gp4 and *Acidobacteria*_Gp17 showed significant negative correlations with pH. In contrast, among the top-ranked dominant species in the five forest types, *Acidobacteria*_Gp4 and *Acidobacteria*_Gp6 were significantly negatively correlated with pH. It can be seen that *Acidobacteria*_Gp4, *Acidobacteria*_Gp17, *Acidobacteria*_Gp1, and *Acidobacteria*_Gp6 are extremely sensitive to pH but do not correlate strongly with other soil physicochemical properties. Acidobacteria is generally able to grow at low pH (pH between 5.5 and 6), but the optimal pH may vary from strain to strain. pH has a direct effect on the permeability of bacterial cell membranes and enzyme activity, which, in turn, affects the growth rate and metabolic activity of Acidobacteria. In a suitably acidic environment, the metabolic activity of Acidobacteria is higher, and they are able to take up nutrients and excrete waste more efficiently. When the environmental pH deviates from the optimal range, it may lead to a decrease in the metabolic activity of the Acidobacteria or even inhibit their growth. Changes in soil nutrients other than soil pH do not have a large impact on the survival of *Acidobacteria*_Gp4, *Acidobacteria*_Gp17, *Acidobacteria*_Gp1, and *Acidobacteria*_Gp6 [56]. This is the reason why *Acidobacteria*_Gp1, *Acidobacteria*_Gp4, and *Acidobacteria*_Gp17 can be keystone species, and *Acidobacteria*_Gp4 and *Acidobacteria*_Gp6 can be dominant species because they are less likely to receive environmental changes. *Acidobacteria*_Gp19 and *Acidobacteria*_Gp21, which are neither dominant nor keystone species, have strong positive correlations with soil TP, AP, and AN. *Acidobacteria*_Gp19 and *Acidobacteria*_Gp21 are more demanding of soil nutrients in terms of nitrogen and phosphorus and can only survive in an environment where the surrounding soil is high in nitrogen and phosphorus nutrients, and, therefore, neither plays a key role in the five forests. In contrast, *Acidobacteria*_Gp1, *Acidobacteria*_Gp4, *Acidobacteria*_Gp17, and *Acidobacteria*_Gp6 are not limited by soil nitrogen content. The strong association between different Acidobacteria and soil nutrients, especially their extreme sensitivity to soil pH, deepens our understanding of soil ecosystems while helping us to develop more effective strategies to combat global climate change.

Mantel analyses showed us that soil TN and pH content are strongly correlated with soil Acidobacteria alpha-diversity (Figure 7). Alpha-diversity is the diversity within a given region or ecosystem that reflects a combination of species richness and evenness. Acidobacteria are known to play an important role in the carbon and nitrogen cycles and other biogeochemical processes in ecosystems [57,58]. Therefore, changes in soil nitrogen content significantly affected the abundance and uniformity of soil Acidobacteria. In addition, Mantel analyses showed a correlation between soil pH content and soil Acidobacteria composition, which may be due to the fact that Acidobacteria are a group of bacteria that are more commonly found in acidic environments and are usually able to grow and reproduce at lower pH values and are, therefore, adapted to acidic environments [58,59,60]. Secondly, pH has an important effect on the growth and metabolic activities of Acidobacteria. Suitable pH helps Acidobacteria maintain the stability of their cellular structure and function properly [11,61,62]. Therefore, both the composition and alpha-diversity of soil Acidobacterial community are related to soil pH. Redundancy analyses also showed that soil pH was progressively enhanced in the *Q. mongolica* and *L. gmelinii* mixed (LGQM) forest, and soil AN was progressively enhanced in the *Betula dahurica* (BD) forest (Figure 8). Soil C/N, TP, and AP were significantly and positively correlated with *Quercus mongolica* (QM) forest, and soil C/N, TP, and AP were significantly and negatively correlated with *Larix gmelinii* (LG) forest and *Betula platyphylla* (BP) forest. This suggested that soil nutrient content (SOC, C/N, AN, TP, AP) is a key environmental factor influencing the composition of soil Acidobacterial communities in different forest types [63,64]. Mantel and RDA analyses revealed the strong association between soil TN, pH, and Acidobacter alpha-diversity and its importance for ecosystem functioning. As participants in key biogeochemical processes, the diversity and community composition of Acidobacteria were significantly affected by soil nutrients and pH, which is an important practical guide for maintaining ecosystem stability and conducting soil nutrient management.

## 5. Conclusions

Our results showed that soil Acidobacterial alpha-diversity of different forest types differed significantly. Meanwhile, there was a significant correlation between soil physico-chemical properties and soil Acidobacterial communities. Soil pH and TN were the main environmental factors impacting soil Acidobacterial communities. Changes in soil physico-chemical properties further affect soil Acidobacterial compositions, ultimately affecting energy flow and nutrient cycling in forest ecosystems. This study further deepens our understanding of the variations in soil Acidobacterial communities in different forest ecosystems and the correlation between soil physico-chemical properties and soil Acidobacterial communities under different forest types. This is essential for the protection of forest ecosystems from the effects of global climate change.

## Figures and Tables

**Figure 1 microorganisms-12-00963-f001:**
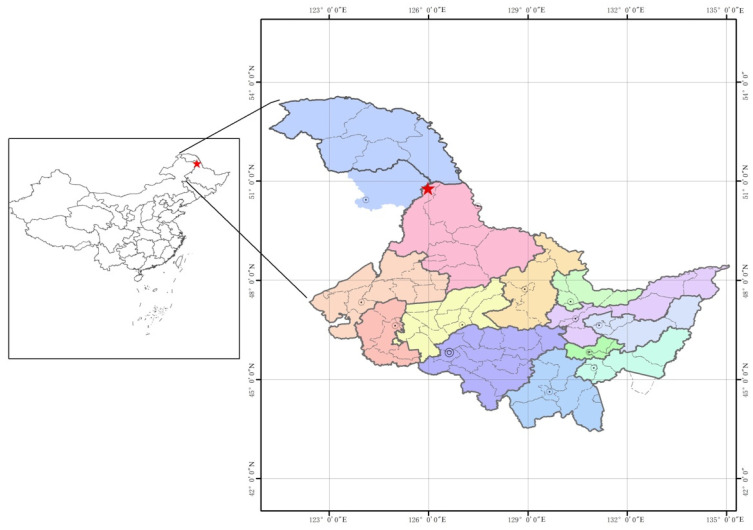
The asterisk indicates the study site in Heilongjiang Province and China.

**Figure 2 microorganisms-12-00963-f002:**
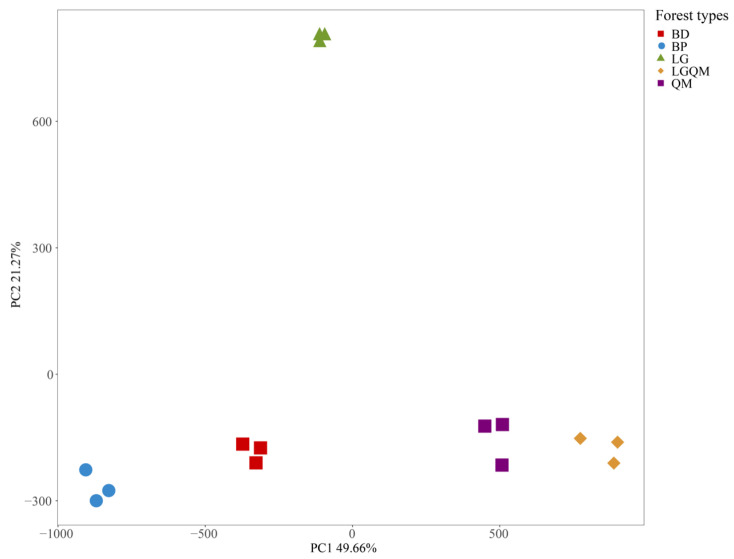
Scores plot of principal component analysis (PCA) showing the values of soil Acidobacteria in different forest types (BD, BP, QM, LGQM, LG). Points of different colors and shapes represent different forest types. The eigenvector is the direction of the axis with the highest variance (most information), called the principal component. Eigenvalues are the variances of a principal component, the relative proportions of which can be interpreted as the variances explained or contribution values. The eigenvalue decreases from the first principal component. The loadings are the eigenvectors multiplied by the square root of the eigenvalues. The loadings are the weight coefficients of the individual raw variables on each principal component. BD: *Betula dahurica* forest; BP: *Betula platyphylla* forest; QM: *Quercus mongolica* forest; LGQM: *Q. mongolica* and *L. gmelinii* mixed forest; LG: *Larix gmelinii* forest.

**Figure 3 microorganisms-12-00963-f003:**
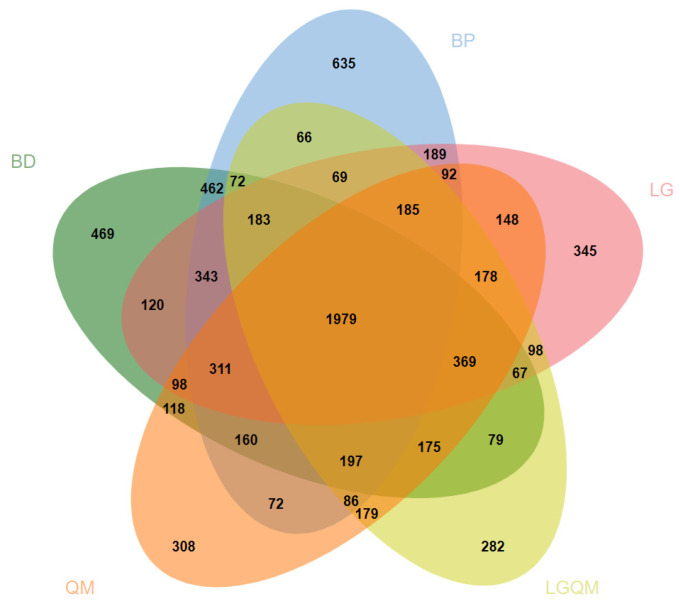
Variance distribution of soil microbial communities under three different treatments (BD, BP, QM, LGQM, LG). Significance levels according to Monte Carlo permutation test (1000 permutations). Different colors represent different treatment conditions. Overlapping areas represent the number of Acidobacterial OTUs common to the different treatment conditions. Non-overlapping areas represent the number of Acidobacterial OTUsgenera specific to the treatment conditions, the number of Acidobacterial OTUst, and the number of genera under different treatment conditions. BD: *Betula dahurica* forest; BP: *Betula platyphylla* forest; QM: *Quercus mongolica* forest; LGQM: *Q. mongolica* and *L. gmelinii* mixed forest; LG: *Larix gmelinii* forest.

**Figure 4 microorganisms-12-00963-f004:**
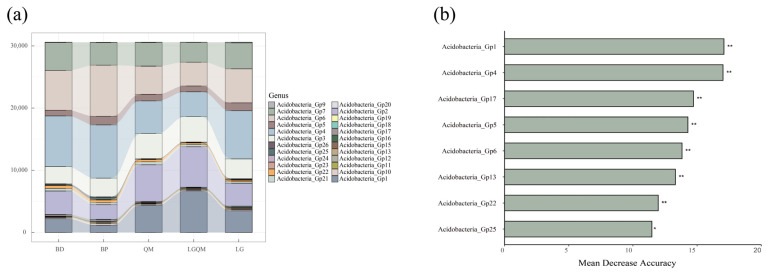
Stacked plot displayed the first 20 Acidobacterial genera in different treatments (**a**), and the Random Forest classification modeling visually shows the dominant Acidobacterial genera in different treatments (**b**). Importance of random forest variables derived from a categorical algorithm in predicting rare Acidobacterial genera in soils under different treatments. The green bars represent variables selected using a categorical algorithm. An asterisk near each bar indicates whether each predictor is significant. BD: *Betula dahurica* forest; BP: *Betula platyphylla* forest; QM: *Quercus mongolica* forest; LGQM: *Q. mongolica* and *L. gmelinii* mixed forest; LG: *Larix gmelinii* forest. * represents significance. ** represents highly significance.

**Figure 5 microorganisms-12-00963-f005:**
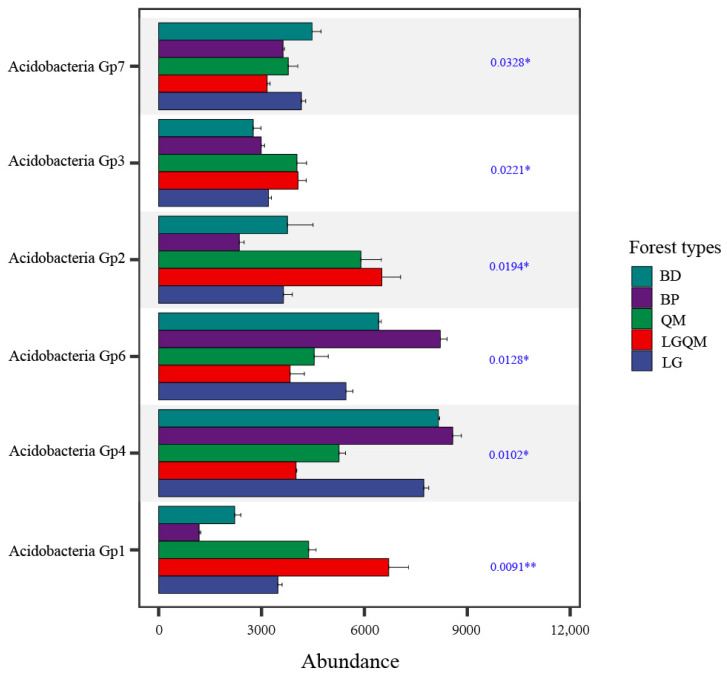
The test of significance plot for differences between groups was used to test for differences in the Acidobacterial genus between treatment groups. The different colored bars represent the different treatment groups, with blue–green representing the BD treatment, purple representing the BP treatment, green representing the QM treatment, red representing the LGQM treatment, and blue representing the LG treatment. Horizontal coordinates represent the percentage of genera in different treatment groups, and vertical coordinates represent different genus names. BD: *Betula dahurica* forest; BP: *Betula platyphylla* forest; QM: *Quercus mongolica* forest; LGQM: *Q. mongolica* and *L. gmelinii* mixed forest; LG: *Larix gmelinii* forest. * represents significance. ** represents highly significance.

**Figure 6 microorganisms-12-00963-f006:**
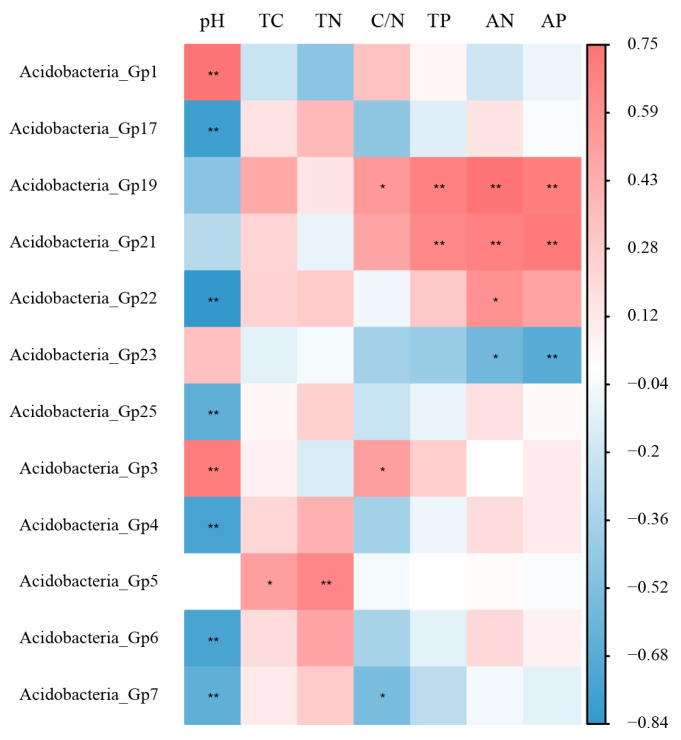
The heatmap of Spearman correlation between Acidobacteria genera and soil physical and chemical properties. The legend on the right indicates the degree of correlation. Red indicates positive correlation, while blue indicates negative correlation. * significance at *p* < 0.05, ** significance at *p* < 0.01.

**Figure 7 microorganisms-12-00963-f007:**
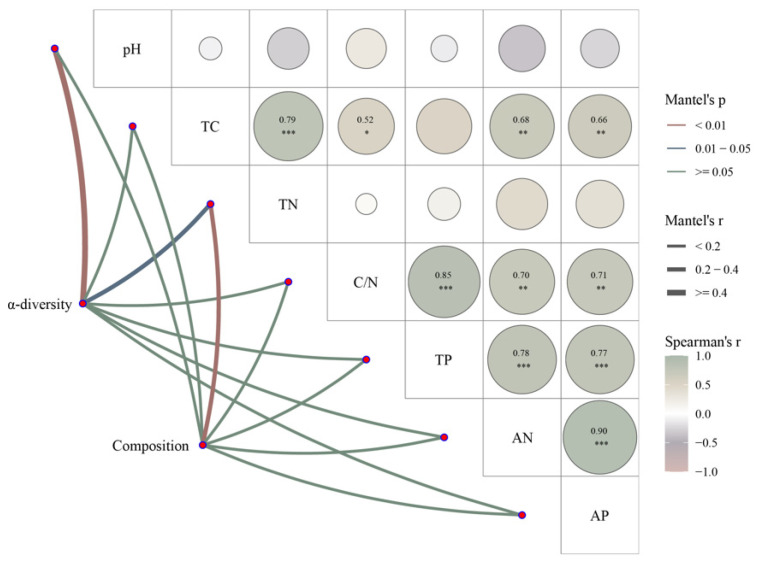
Mantel analysis was used to clarify the relationship between soil Acidobacterial community’s alpha-diversity and composition with soil physicochemical properties. The red and blue lines represent different levels of correlation, and the green line represents no correlation. The thickness of the line (Spearman’s correlation coefficients) represents the magnitude of the correlation. The thicker the line, the greater the correlation. The thinner the line, the smaller the correlation. pH: Pondus hydrogenii; TN: Total Carbon; TN: Total nitrogen; C/N: carbon to nitrogen ratio; TP: Total phosphorus; AN: Available nitrogen; AP: Available phosphorus. Composition was derived based on principal component analysis (PCA). * represents significance. ** denotes highly significance. *** represents extremely significance.

**Figure 8 microorganisms-12-00963-f008:**
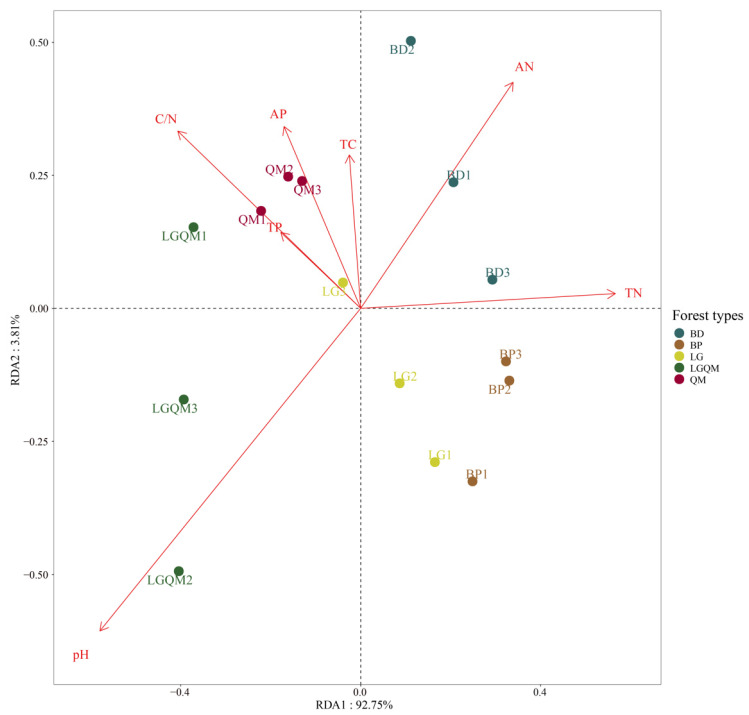
Redundancy analysis (RDA) combines correspondence analysis with multiple regression analysis, where each step of the calculation is regressed against environmental factors. Dots: each dot represents a sample; different colored dots belong to different subgroups (BD, BP, QM, LGQM, LG); the closer the distance between two dots, the higher the functional similarity of the two samples. Arrows represent different influencing factors; the angle between the influencing factors represents the magnitude of correlation between them. Acute angle indicates that the two factors are positively correlated. Right angle indicates that the two factors are not correlated. Obtuse angle indicates that the two factors are negatively correlated. The length of the ray: the longer the ray, the greater the influence of the factor on the structure and function of the colony; the angle between the arrow ray and the coordinate axis represents the size of the correlation between a certain environmental factor and the coordinate axis; the smaller the angle, the higher the correlation; the position of the sample projection point on the blue arrow: an approximate representation of the size of the value of the factor in the corresponding sample. Percentage next to the axes represents the proportion of the variance in the raw data that can be explained by the corresponding axes. BD: *Betula dahurica* forest; BP: *Betula platyphylla* forest; QM: *Quercus mongolica* forest; LGQM: *Q. mongolica* and *L. gmelinii* mixed forest; LG: *Larix gmelinii* forest.

**Table 1 microorganisms-12-00963-t001:** Soil physicochemical properties in different forest types (BD, BP, QM, LGQM, LG).

Variable	BD	BP	QM	LGQM	LG
pH	4.61 ± 0.39 c	5.27 ± 0.06 b	5.43 ± 0.15 ab	5.86 ± 0.34 a	5.49 ± 0.13 ab
TC (g·kg^−1^)	53.97 ± 3.53 a	76.17 ± 2.04 b	108.40 ± 0.00 a	49.87 ± 3.95 a	60.28 ± 3.48 c
TN (g·kg^−1^)	3.47 ± 0.34 b	4.27 ± 0.15 a	4.20 ± 0.10 a	2.76 ± 0.13 c	4.12 ± 0.13 a
C/N	15.59 ± 0.76 c	17.86 ± 0.35 b	25.82 ± 0.61 a	18.07 ± 1.58 b	14.66 ± 1.29 c
TP (g·kg^−1^)	2.07 ± 0.06 bc	2.33 ± 0.15 ab	2.60 ± 0.20 a	2.17 ± 0.29 b	1.77 ± 0.15 c
AN (mg·kg^−1^)	78.67 ± 2.96 b	83.75 ± 2.22 a	87.34 ± 1.59 a	38.59 ± 2.18 c	24.63 ± 2.60 d
AP (mg·kg^−1^)	36.17 ± 1.23 b	37.28 ± 0.42 b	41.25 ± 1.40 a	35.86 ± 0.00 b	27.43 ± 0.81 c

Note: BD: *Betula dahurica* forest; BP: *Betula platyphylla* forest; QM: *Quercus mongolica* forest; LGQM: *Q. mongolica* and *L. gmelinii* mixed forest; LG: *Larix gmelinii* forest. Six replications were performed for each treatment. The data show the mean, variance, and stderr of different treatment groups (*p* < 0.05). The table displays the *p*-values and fdr. pH: Pondus hydrogenii; TC: Total Carbon; TN: Total nitrogen; C/N: carbon to nitrogen ratio; TP: Total phosphorus; AN: Available nitrogen; AP: Available phosphorus. Letters (a–d) represent the significance of soil physico-chemical properties among different forest types. The same letters represent no significance and different letters represent significance.

**Table 2 microorganisms-12-00963-t002:** Alpha diversity of fungal communities under different treatments (BD, BP, QM, LGQM, LG).

Forest Type	Richness	Chao1	ACE	Shannon	Evenness
BD	3687.3 ± 154.1 a	4444.1 ± 159.8 a	4591.5 ± 127.1 a	10.1 ± 0.0 a	0.9 ± 0.0 a
BP	3661.6 ± 71.9 a	4412.6 ± 14.9 a	4568.6 ± 31.0 a	10.2 ± 0.0 a	0.9 ± 0.0 a
QM	3235.6 ± 94.7 c	3896.5 ± 104.2 c	4010.1 ± 146.9 c	9.9 ± 0.0 b	0.9 ± 0.0 b
LGQM	3055.0 ± 31.7 c	3749.3 ± 43.3 c	3876.7 ± 60.7 c	9.7 ± 0.0 c	0.8 ± 0.0 c
LG	3436.6 ± 103.5 b	4164.5 ± 18.1 b	4284.3 ± 43.5 b	9.9 ± 0.0 b	0.9 ± 0.0 b

Note: BD: *Betula dahurica* forest; BP: *Betula platyphylla* forest; QM: *Quercus mongolica* forest; LGQM: *Q. mongolica* and *L. gmelinii* mixed forest; LG: *Larix gmelinii* forest. Six replications were performed for each treatment. The data are expressed as the mean ± standard deviation; lowercase letters indicate significant differences (*p* < 0.05).

## Data Availability

Data are contained within this article.

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
