# Peer review of "Diversity and Composition of Soil Acidobacterial Communities in Different Temperate Forest Types of Northeast China"

_microorganisms, 2024, doi:10.3390/microorganisms12050963_

Round 1

Reviewer 1 Report

Comments and Suggestions for Authors

Abstract - what new knowledge for the international scientific community comes out of this research ? How much is a research report of local interest and how much something that has novel aspects and relevance to other researchers from other scientific centers ? In my opinion, the authors themselves diminish the importance of this research by emphasizing that it is a case study.

Introduction - there is a lack of good demonstration of knowledge gaps and demonstration of the general importance of this type of research to increase the current state of knowledge. The more it relates to a case study, the fewer citations in the future. What universal the authors want to show and how it will affect the current state of knowledge ? Please note that through a case study the authors can show universal mechanisms relevant to everyone. The more it is a case study, the less interest many researchers will have. It is a matter of appropriate description.

Materials and methods - can the experiments be verified on the basis of the present description ? In my opinion, the description should be more precise.

Please check the record of results. E.g. " 3687.3±154.08a 4444.1±159.81a " - is it possible for the error to be more precise than the measurement result itself ? No - all numbers must have the same number of decimal places. Please correct throughout the manuscript.

The description of the results is generally OK, but please note that many elements are repeated in the discussion. For me, what is missing is a discussion showing some trends of interest to other readers. Please show what global relevance this has, e.g. to other similar places in the world. The message must be universal, and then the manuscript will be very good. Here the authors need to rewrite the text more strongly.

Comments on the Quality of English Language

OK, some small changes would make the manuscript perfect

Author Response

Response to Reviewer 1 Comments

Dear Editor Sylvia Shi,

Dear Reviewer 1,

Thank you very much for giving us the opportunity to revise our manuscript entitled “Diversity and Composition of Soil Acidobacterial Communities in Different Temperate Forest Types of Northeast China(microorganisms-2969910)”. Thank you very much for your valuable comments on our manuscript. According to your comments and suggestions, we carefully revised throughout our manuscript.

Comments:

Comments and Suggestions for Authors :

Q1:

Abstract - what new knowledge for the international scientific community comes out of this research ? How much is a research report of local interest and how much something that has novel aspects and relevance to other researchers from other scientific centers ? In my opinion, the authors themselves diminish the importance of this research by emphasizing that it is a case study.

Response: 

Dear reviewer,

Thank you for your constructive comments for our manuscript. We agree with your comments. According to your suggestion, we have revised Abstract. Please see the lines 42-49.

Comments:

Comments and Suggestions for Authors :

Q1:

Introduction - there is a lack of good demonstration of knowledge gaps and demonstration of the general importance of this type of research to increase the current state of knowledge. The more it relates to a case study, the fewer citations in the future. What universal the authors want to show and how it will affect the current state of knowledge ? Please note that through a case study the authors can show universal mechanisms relevant to everyone. The more it is a case study, the less interest many researchers will have. It is a matter of appropriate description.

Response: 

Dear reviewer,

Thank you for your constructive comments for our manuscript. We agree with your comments. According to your suggestion, we have revised this description. Please see the lines 68-90.

Comments:

Comments and Suggestions for Authors :

Q1:

Materials and methods - can the experiments be verified on the basis of the present description ? In my opinion, the description should be more precise.

Response: 

Dear reviewer,

Thank you for your constructive comments for our manuscript. We agree with your comments. According to your suggestion, we have revised this description. Please see the lines 138-154.

Comments:

Comments and Suggestions for Authors :

Q1:

Please check the record of results. E.g. " 3687.3±154.08a 4444.1±159.81a " - is it possible for the error to be more precise than the measurement result itself ? No - all numbers must have the same number of decimal places. Please correct throughout the manuscript.

Response: 

Dear reviewer,

Thank you for your constructive comments for our manuscript. We agree with your comments. According to your suggestion, we have revised this description. Please see the line 218.

Comments:

Comments and Suggestions for Authors :

Q1:

The description of the results is generally OK, but please note that many elements are repeated in the discussion. For me, what is missing is a discussion showing some trends of interest to other readers. Please show what global relevance this has, e.g. to other similar places in the world. The message must be universal, and then the manuscript will be very good. Here the authors need to rewrite the text more strongly.

Response: 

Dear reviewer,

Thank you for your constructive comments for our manuscript. We agree with your comments. According to your suggestion, we have revised this description. Please see the lines 345-513.

Comments:

Comments and Suggestions for Authors :

Q1:

Comments on the Quality of English Language. OK, some small changes would make the manuscript perfect.

Response: 

Dear reviewer,

Thank you for your constructive comments for our manuscript. We agree with your comments. According to your suggestion, we revised the English expression throughout the text.

Reviewer 2 Report

Comments and Suggestions for Authors

Overall, the work is well-structured, and this contribution should be considered for publication after addressing the following comments.

1.      Graphical Abstract should be prepared to make the manuscript more attractive to Potential readers.

2.      Re-write and organize the abstract into distinct sections such as objectives, methodology, results, and conclusions. Also, correct Through ANOVA(Analysis of variance) ….

3.      In Material method 2.2 must be re-write  in soil sampling  five different forest types ….. completed and re-write then first the unreinforced layer be removed how to remove completely

4.      In section 2.3 Soil pH testing using a soil pH meter (write the model, detail)

5.      In discussion line no 422 Mantel analyses, please write some line about Mantel analysis

6.      Line 429 e 430 Acidobacteria are a group of bacteria that are more commonly found in acidic environments, and are usually able to grow and reproduce at lower pH values, how much is the pH values? And how it affects the metabolic activity of Acido bacteria

Author Response

Response to Reviewer 2 Comments

Dear Editor Sylvia Shi,

Dear Reviewer 2,

Thank you very much for giving us the opportunity to revise our manuscript entitled “Diversity and Composition of Soil Acidobacterial Communities in Different Temperate Forest Types of Northeast China(microorganisms-2969910)”. Thank you very much for your valuable comments on our manuscript. According to your comments and suggestions, we carefully revised throughout our manuscript.

Comments:

Comments and Suggestions for Authors :

Q1:

  1. Graphical Abstract should be prepared to make the manuscript more attractive to Potential readers.

Response: 

Dear reviewer,

Thank you for your constructive comments for our manuscript. We agree with your comments. According to your suggestion, we have made a Graphical Abstract.

Comments:

Comments and Suggestions for Authors :

Q1:

  1. Re-write and organize the abstract into distinct sections such as objectives, methodology, results, and conclusions. Also, correct Through ANOVA(Analysis of variance) …

Response: 

Dear reviewer,

Thank you for your constructive comments for our manuscript. We agree with your comments. According to your suggestion, we have revised Abstract. Please see the lines 15-49.

Comments:

Comments and Suggestions for Authors :

Q1:

  1. In Material method 2.2 must be re-write  in soil sampling  five different forest types ….. completed and re-write then first the unreinforced layer be removed how to remove completely.

Response: 

Dear reviewer,

Thank you for your constructive comments for our manuscript. We agree with your comments. According to your suggestion, we have revised this description. Please see the lines 122-131.

Comments:

Comments and Suggestions for Authors :

Q1:

  1. In section 2.3 Soil pH testing using a soil pH meter (write the model, detail.

Response: 

Dear reviewer,

Thank you for your constructive comments for our manuscript. We agree with your comments. According to your suggestion, we have revised this description. Please see the lines 138-142.

Comments:

Comments and Suggestions for Authors :

Q1:

  1. In discussion line no 422 Mantel analyses, please write some line about Mantel analysis.

Response: 

Dear reviewer,

Thank you for your constructive comments for our manuscript. We agree with your comments. According to your suggestion, we have revised this description. Please see the line 457.

Comments:

Comments and Suggestions for Authors :

Q1:

  1. Line 429 e 430 Acidobacteria are a group of bacteria that are more commonly found in acidic environments, and are usually able to grow and reproduce at lower pH values, how much is the pH values? And how it affects the metabolic activity of Acido bacteria.

Response: 

Dear reviewer,

Thank you for your constructive comments for our manuscript. We agree with your comments. According to your suggestion, we have revised this description. Please see the lines 465-472.

Round 2

Reviewer 1 Report

Comments and Suggestions for Authors

The authors have revised the manuscript. It is difficult to provide more insightful changes in such a fast time, so the quality of the presentation has increased, but still not changes that dramatically increase the overall quality of the scientific communication. 

Author Response

Dear Reviewer 1,

Thank you very much for giving us the opportunity to revise our manuscript entitled “Diversity and Composition of Soil Acidobacterial Communities in Different Temperate Forest Types of Northeast China(microorganisms-2969910)”. Thank you very much for your valuable comments on our manuscript. According to your comments and suggestions, we carefully revised throughout our manuscript.

Comment:

The authors have revised the manuscript. It is difficult to provide more insightful changes in such a fast time, so the quality of the presentation has increased, but still not changes that dramatically increase the overall quality of the scientific communication.

Response:

Dear reviewer:

Thank you for your considerable comment for our manuscript. We have revised our entire manuscript according to your comment. Please see the revision manuscript.
